# Distribution of Wheat-Infecting Viruses and Genetic Variability of Wheat Streak Mosaic Virus and Barley Stripe Mosaic Virus in Kazakhstan

**DOI:** 10.3390/v16010096

**Published:** 2024-01-08

**Authors:** Anastasiya Kapytina, Mariya Kolchenko, Nazym Kerimbek, Alexandr S. Pozharskiy, Gulnaz Nizamdinova, Aisha Taskuzhina, Kamila Adilbayeva, Marina Khusnitdinova, Malika Amidullayeva, Ruslan Moisseyev, Zulfiya Kachiyeva, Dilyara Gritsenko

**Affiliations:** 1Laboratory of Molecular Biology, Institute of Plant Biology and Biotechnology, Almaty 050040, Kazakhstan; anastasiya.kapytina@mail.ru (A.K.); nazym_kerimbek@mail.ru (N.K.); mamidullaeva@gmail.com (M.A.); kachieva@gmail.com (Z.K.); 2Department of Molecular Biology and Genetics, Al Farabi Kazakh National University, Almaty 050040, Kazakhstan; 3Research Institute of Applied and Fundamental Medicine, Kazakh National Medical University, Almaty 050000, Kazakhstan

**Keywords:** WSMV, BSMV, coat protein, γB protein

## Abstract

Wheat is an essential cereal crop for the economy and food safety of Kazakhstan. In the present work, a screening of wheat and barley from different regions of Kazakhstan was conducted using newly developed specific primers for reverse transcription PCR and loop-mediated isothermal amplification (LAMP) assays. In total, 82 and 19 of 256 samples of wheat and barley tested positive for wheat streak mosaic virus (WSMV) and barley stripe mosaic virus (BSMV), respectively. A phylogenetic analysis using two independent methods revealed that most of the analyzed isolates had a European origin. Molecular data on the distribution and diversity of cereal viruses in Kazakhstan were obtained for the first time and will help lay a foundation for the implementation of genetics and genomics in wheat phyto-epidemiology in the country.

## 1. Introduction

Wheat (*Triticum aestivum* L.) is a crop essential to the world in general and Kazakhstan in particular, where it is important both for export and domestic consumption [1]. In 2021, its harvest area reached 12.7 million hectares, with an average yield of 1.1 tons per hectare, resulting in a total production of 14 million tons of grain [2]. Plant viruses pose a significant threat because their spread is difficult and costly to control, and there is no reliable treatment aside from the total eradication of suspected vectors and infected plants [3]. Viruses of special importance to wheat include barley stripe mosaic virus (BSMV), wheat streak mosaic virus (WSMV), triticum mosaic virus (TriMV), and *Emaravirus tritici* (previously HPWMoV) [4,5,6]. BSMV and wheat streak mosaic (WSM) complex, comprising the latter three viruses, have similar symptoms, such as a mosaic pattern of yellow or light green stripes running parallel to the veins of the infected leaves (Figure 1B), and their combined infection results in disease synergism detectable via increased titers of all viruses [7]. Grain yield is shown to decline exponentially from WSM infections [8], and some outbreaks lead to a total crop failure [9]. Wheat dwarf virus, which is currently gaining prominence in Europe, can also cause streak-like leaf chlorosis [10], but the dwarfism associated with it has not been reported in Kazakhstan or its neighboring regions.

Barley stripe mosaic virus (BSMV), belonging to the genus *Hordeivirus*, family Virgaviridae, can infect barley, wheat, and oats [11,12,13]. Its genome consists of three positive-sense single-stranded RNA subunits, RNAα, RNAβ, and RNAγ, collectively encoding seven polypeptides: αa, βa (CP), ßb (TGB1), ßc (TGB3), ßd (TGB2), γa, and γb [14,15,16]. The latter, cysteine-rich 17 kDa γb protein, tends to accumulate during the BSMV infection cycle and plays a pivotal role in pathogenesis [14,17], systemic movement [18], and silencing suppression [19,20]. BSMV has no natural transmitting vectors and is distributed via pollen or seeds [21]. Its uncontrolled spread may lead to up to 75% yield losses [22,23], but it can be prevented through the use of certified virus-free planting material [12].

Wheat streak mosaic virus (WSMV), of the genus *Tritimovirus*, family Potyviridae, is a highly destructive pathogen affecting a wide range of cereal crops and significantly impacting worldwide grain production [6]. It has a rod-shaped virion containing a ssRNA (+) genome of 9.3–9.4 kb, which has one open reading frame and encodes a large polypeptide from which 10 mature proteins are cleaved (P1, HC-Pro, P3, 6K1, CI, 6K2, VPg, NIa, NIb, and CP) [24,25,26]. Among them are the nuclear inclusion b cistron (NIb) protein (57 kDa), a replicase strongly conserved among potyviruses [27,28], and the coat protein (CP, 37 kDa), which, besides virion formation, is involved in cell-to-cell movement via its C-terminal aspartic acid residues [6,29]. WSMV has been detected in all major grain exporter regions, including the USA [30], Mexico [31], Argentina [32], Russia [33], Iran [34,35], and Australia [36,37], and has the potential to jeopardize their production rates.

Triticum mosaic virus (TriMV) is a ssRNA (+) virus with a genome of about 10.2 kb, which is translated into a single polyprotein of approximately 350 kDa [27,38,39], which is cleaved into several proteins characteristic for the Potyviridae family that show functional similarity to those of WSMV [40,41]. TriMV is mostly found in co-infection with WSMV, where their synergetic interaction leads to increased virus titers and yield losses [42,43].

HPWMoV (*Emaravirus tritici* [44], previously High Plains wheat mosaic emaravirus) has an octopartite, ssRNA (−) genome that is enveloped by a 32 kDa nucleocapsid (NC) encoded by RNA3 [45,46,47]. Its RNA1 and RNA2 encode an RNA-dependent RNA polymerase (RdRp) and a glycoprotein precursor protein, respectively, and P4-P6 are homologous to other *Emaravirus* proteins, unlike the gene products of RNA7 and RNA8 [45]. HPWMoV not only causes significant losses in the USA [48], but also infects wheat crops in Europe [49] and Australia [50].

The natural vector for WSMV, TriMV, and HPWMoV is *Aceria tosichella* Keifer [51,52]. The three viruses can be transmitted either individually or in various combinations, with yield losses exacerbated by co-infection [42]. Studies of wheat fields in the Great Plains have also shown that WSMV and HPWMoV are primarily detected as single infections, while TriMV is mostly found together with WSMV [53], and even collective triple infections are not uncommon [48]. Seed transmission is less prominent for the WSM complex, with Jones et al. citing 0.5% to 1.5% [54], while a recent study found WSMV in 13% of seeds from infected susceptible cultivars [55].

The detection of WSMV and BSMV is commonly conducted via commercial serological kits based on triple or double antibody sandwich ELISA, or molecular methods. Multiplex RT-PCR has been successfully applied for the screening and quantification of WSMV and TriMV [56,57], and RT-qPCR has been shown to be twice and thrice more sensitive than ELISA in the detection of TriMV and HPWMoV, respectively [58]. For BSMV, Zarzyńska et al. demonstrated the limitations of ELISA as a diagnostic tool: it could not detect mild infections with moderate virus titers, while RT-PCR [59] and LAMP [60] were more sensitive. RT-qPCR is a valuable tool in studying virus gene expression and virus–host interactions [6].

Although no systematic studies of wheat viruses have been conducted in Kazakhstan to date, there are unpublished data reporting typical symptoms of viral disease in wheat and barley observed in different areas around the country. Therefore, considering the particular importance of wheat and other cereals for the economy and food safety of Kazakhstan, a comprehensive study of the distribution of wheat viruses involving molecular methods of identification and analysis is necessary for disease management and the prevention of massive outbreaks in the country.

In the present work, we evaluated the performance of established primers and designed new specific sets for the detection of WSMV and BSMV, which were tested on wheat and barley from different regions of the country. Amplicons produced by the new primers were sequenced and aligned with the accessions available in GenBank. A phylogenetic analysis was performed to shed new light on the genetic variability of WSMV and BSMV at local and global levels and to elucidate the origin, distribution, and impact of the viruses on wheat culture in Kazakhstan. The identification and population studies of WSMV and BSMV were conducted for the first time in Kazakhstan and will become the basis for further investigations on wheat viruses and protection against them.

## 2. Materials and Methods

### 2.1. Sample Collection and RNA Extraction

In 2021, 20 wheat and barley fields in northern and southeastern Kazakhstan were surveyed for plants displaying chlorosis, streaks, and mosaic (Figure 1A,B). In total, 256 leaf samples of both symptomatic and asymptomatic plants were collected (Table A1). The GPS coordinates of the sampling sites were recorded with a portable GPS navigator (Garmin 66S) and mapped using ArcGIS 10.5 software (Figure 1A). The plants were transported in plastic bags, frozen in liquid nitrogen, and stored at −80 °C until further use.

RNA was extracted using cetyltrimethylammonium bromide (CTAB) [61]. To confirm the quality of the total RNA extracts, 200 ng of RNA was separated on a 2% (*w*/*v*) agarose gel. Complementary DNA (cDNA) was synthesized using RevertAid Reverse Transcriptase (Thermo Fisher Scientific, Waltham, MA, USA). After combining 200 ng of RNA, 0.5 μg Oligo-dT, and 0.5 μg random hexamer primers in a final volume of 15 μL, the mix was denatured for 10 min at 72 °C and then cooled on ice. After adding 5× RT reaction buffer, 0.5 mM dNTPs, and 100 U reverse transcriptase, cDNA was synthesized for 1 h at 45 °C. The integrity of the isolated RNA was confirmed on agarose gel.

**Figure 1 viruses-16-00096-f001:**
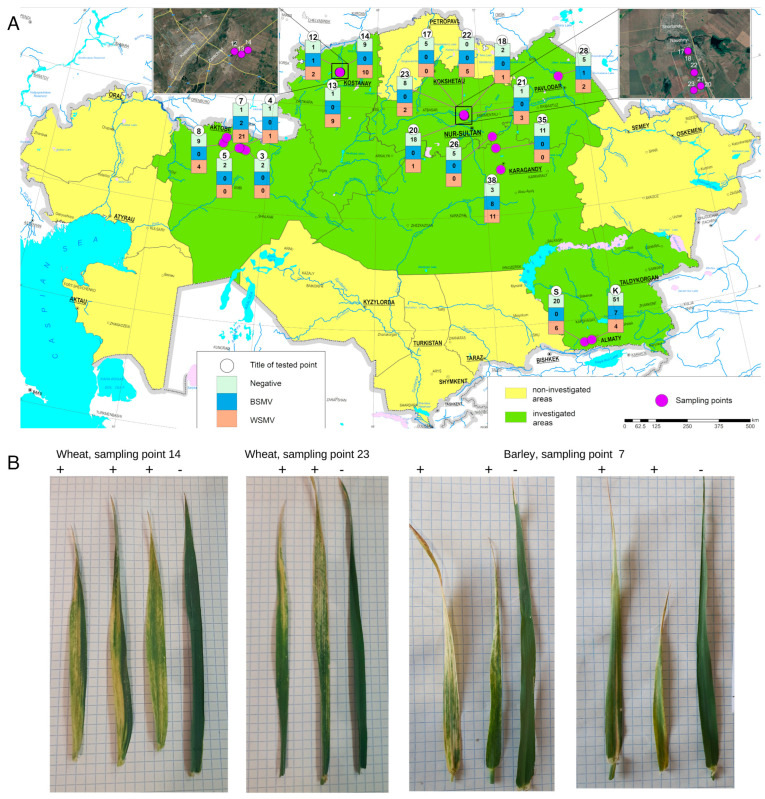
Sampling of wheat and barley in the northern and southeastern regions of Kazakhstan. (**A**) Map of Kazakhstan with sampling sites; location points according to Table A1. (**B**) Examples of wheat and barley leaves with symptoms of viral disease; all leaves are shown alongside a healthy leaf (rightmost leaf in each photo); grid size: 5 mm.

### 2.2. Primer Design

The initial screening was performed using previously established detection primers (Table 1): two sets of primers for WSMV [43,62], and one of each for BSMV [59], TriMV [43], and HPWMoV [43]. However, the majority of samples showing strong symptoms of viral infection were negative for all viruses, or produced amplicons outside of target sizes (Appendix A). Therefore, in the current work, we developed and tested new sets of primers for WSMV and BSMV to detect strains and isolates circulating in Kazakhstan with possible mutations in the binding sites of previously established primers.

RT-PCR primers for each virus were designed using Primer3 based on all the sequences of the WSMV coat protein and BSMV γB protein available in NCBI and described in Appendix A, and the specificity of each primer set was tested in silico using Primer-BLAST (https://www.ncbi.nlm.nih.gov/ (accessed on 5 January 2024)) (Table 1).

Three sets of LAMP primers targeting conservative regions of WSMV Nlb and CP encoding genes were designed using the Neb LAMP primer design tool (https://lamp.neb.com/#!/ (accessed on 5 January 2024)) (Table 2). The analysis of all Nlb and CP sequences available in NCBI and the determination of conservative regions was performed in UGENE [63]. LAMP primers were selected based on the values of free energy at the 3′ ends of F2/B2, F3/B3, and LF/LB, and the 5′ end of F1c/B1c, set to –4 kcal/mol or less. Because the 5′ end of F1c after amplification corresponded to the 3′ end of F1, stability was very important (https://primerexplorer.jp/e/ (accessed on 5 January 2024)). Additionally, loop primers for each set were developed to reduce the amplification time and improve the specificity.

### 2.3. Detection of Viruses Using PCR and LAMP

PCR screening for WSMV, BSMV, TriMV, and HPWMoV was performed according to the protocols described in the respective works [43,59,62]. For the newly developed primers, PCR was conducted in 25 μL of reaction mix containing 10 mM of each primer and 2 μL of cDNA as a template for amplification. The PCR cycling conditions were 95 °C for 3 min, followed by 30 cycles of 30 s at 95 °C, 20 s at 54 °C (for WSMV) and 58 °C (for BSMV), and 40 s at 72 °C. After a final extension of 5 min at 72 °C, 10 μL of each PCR reaction was separated on a 2.0% (*w*/*v*) agarose gel.

The detection of WSMV using LAMP was performed in accordance with the protocol for the WarmStart Colorimetric LAMP 2× Master Mix with UDG (M1804L, New England BioLabs, Ipswich, MA, USA). The primer mix was prepared separately for each set, including LB/RB or both, and 1 μL of sample RNA was used as a template. The resulting mix was amplified for 30–60 min at 65 °C. The results were analyzed both visually (phenol red indicator) and on 1.5% agarose gel.

### 2.4. Targeted Sequencing

Short amplicons produced via PCR (Table 1) and almost complete sequences of WSMV coat protein gene and BSMV γB gene amplified using a different set of primers (Table 3) were sequenced for the investigation of genetic diversity. The 950 bp sequences of WSMV CP and 460 bp sequences of BSMV γB were amplified using the overlapping method [64]. The sequencing reaction was performed in 10 μL of mix containing 5–10 ng of the PCR product, 3.2 mM of either the forward or reverse primer for each virus, 1 μL BigDye™ Terminator Reaction Mix (Thermo Fisher Scientific, Waltham, MA, USA), and 1.5 μL BigDye™ Terminator sequencing buffer. The cycling conditions were 96 °C for 1 min, followed by 25 cycles of 10 s at 96 °C, 5 s at 50 °C, and 4 min at 60 °C. Sequencing was carried out on a 3500 Genetic Analyzer (Applied Biosystems, Waltham, MA, USA) using the StdSeq50_POP7 run mode.

All sequences were assembled using DNAMAN and deposited in the NCBI GenBank (OP793649–OP793678).

### 2.5. Phylogenetic Analysis

The nucleotide sequences coding for the WSMV CP region and BSMV γB protein were subsequently aligned with relevant accessions from the NCBI GenBank (Appendix A) using MAFFT [65], including whole genome sequences of both viruses, the polyprotein or coat protein cDNA of WSMV, and the γRNA or γB protein of BSMV (Appendix A). The alignment was inspected and refined manually to exclude low-quality sequences and to correct alignment mistakes. Phylogenetic trees were constructed in MrBayes (average standard deviation of split frequencies < 0.01) [66,67].

The combined set of sequences was tested for possible recombination events using RDP5 software [68] with the following methods: RDP [69], GENECONV [70], Bootscan [71], Maxchi [72], Chimaera [73], SiSscan [74], and 3Seq [75].

## 3. Results

### 3.1. RT-PCR and LAMP Detection of Wheat Viruses

Sample screening for WSMV, BSMV TriMV, and HPWMoV using pre-established detection primers delivered mixed results. TriMV and HPWMoV were not detected, while the two sets of WSMV primers amplified sequences of various lengths from different isolates. One set of primers by Byamukama et al. detected WSMV in 10 of the isolates, while the expected 750-bp fragment of the set WS-8166-8909 [62] was amplified in 32 of them. As for BSMV, eight specimens were positive for the TGB2 set [59].

The newly developed primer WSMV-8156-8783 identified WMSV infection in 82 samples, 24 of which were previously detected via WS-8166-8909 and 3 via WSMV, and 4 isolates amplified all three primers (Figure 2). However, four and three were detected only by WS-8166-8909 and WSMV, respectively. Cumulatively, the new primer set made it possible to identify 51 more isolates, the identity of which was confirmed via sequencing.

BSMV-2763-2868 produced amplicons in 19 isolates, allowing to identify 13 new infections. A mixed infection of WSMV and BSMV was detected in 10 samples.

Three sets of LAMP primers complementary to WSMV CP and Nlb protein coding regions were designed (Table 3) and tested on the 82 isolates where WSMV was detected. Sets No. 1 and No. 3 indicated nonspecific amplification in some samples and the negative control, so set No. 2 was chosen as the best suitable primer set because it showed higher specificity and sensitivity (Appendix A). The positive LAMP products were analyzed using the colorimetric method due to the pH-sensitive indicator phenol red changing from red to yellow, as well as gel electrophoresis displaying multiple bands of different sizes due to the formation of stem-loop DNA of different lengths (Figure 3).

The newly developed primers for RT-PCR and LAMP were consistently amplified in the same samples.

### 3.2. Genetic Variability of WSMV and BSMV

To confirm the sequence identity between the amplification products and the corresponding genomic region of WSMV or BSMV, both amplicons and extended sequences of the target regions were sequenced and aligned with the accessions retrieved from the NCBI GenBank (Appendix A).

From the positive sample pool, 23 WSMV and 6 BSMV isolates from different regions of Kazakhstan were randomly selected, sequenced, and subjected to phylogenetic analysis. The 950 bp of WSMV CP (Figure 4), as well as 460 bp of BSMV γB (Figure 5) coding regions were used for analysis. Additionally, cladograms based on the amplicons (274 bp for WSMV and 134 bp for BSMV) were constructed (Appendix A).

Almost all Kazakh WSMV isolates formed a clade close to the isolates of European origin (Ukraine, Czech Republic, and Hungary). For isolate KZ57, both sequences had a similar distinct position in all cladograms and were located near Iranian and American isolates.

The sequenced fragments of BSMV demonstrated relative proximity to European isolates (Figure 5). However, the low number of foreign isolates available for comparison limited the possible conclusions about the clustering on the global scale. All sequenced isolates were detected on the “Svetlana” farm near Almaty (location point “S”), except for KZ8 (Pavlodar, location point 28). The latter formed a well-supported (bootstrap value 99) clade with the isolates from Egypt and China.

Overall, the phylogenetic trees constructed using short (Appendix A) and long reads of WSMV and BSMV displayed relative similarity between the composition of larger groupings such as continental division, although short sequences could not be used to distinguish the phylogenetic relationships within them. This indicates that short sequences, instead of long ones, can be used for the rapid evaluation of clade numbers in the populations, which reduces the time required for workflow and analysis. However, the loss of complexity between the topologies of the two BSMV trees is quite severe, and the one based on shorter sequences retains only one distinct clade (American).

The RDP5 software test revealed a single potential recombination event involving WSMV sample KZ28 from ‘Lyubava’ farm and two Ukrainian isolates (MH523357.1 and MH523356.1) with Bootscan (*p*-value 1.44 × 10^–4^), Maxchi (*p*-value 2.3 × 10^–6^), and 3Seq (*p*-value 5.85 × 10^–13^); however, the exact breakpoint positions and relations between the relevant sequences could not be determined due to the incompleteness of data, so we consider this unreliable. All other potential recombination events were not supported by any of the used tests, and/or were identified as false positive.

## 4. Discussion

The early detection of plant viruses using molecular methods is essential for implementing quarantine measures to prevent further disease spread. Therefore, sensitive, quick, and robust molecular methods, such as LAMP, RT-PCR, RT-qPCR, and dPCR will always be relevant and necessary [76]. The natural evolution of viruses and rapidly expanding genetic data enable the development of new, more reliable detection systems and the modification of existing ones. In this work, the previously established primer sets were incapable of detecting WSMV and BSMV in several samples with strong symptoms of viral infection, possibly due to mutations in the binding sites of the primers on behalf of the viral isolates circulating in Kazakhstan. The primers developed in the present work allowed us to identify 82 and 19 instances of WSMV and BSMV infections, respectively, and to perform a comparative phylogenetic analysis. Additionally, we were able to demonstrate the successful amplification of WSMV within 60 min at 65 °C using an RT-LAMP assay.

TriMV and HPWMoV were not detected in the collected samples of wheat and barley. This may have occurred because of the specificity of the primers used to detect these viruses, the low concentration of viruses in plants, inappropriate time for testing, dependence on the distribution of natural vectors, or the absence of viruses. TriMV was not detected outside North America before 2023 [77], and its limited distribution in Iraq suggests a recent introduction. So far, Ukraine [78] and Iran [79] report the presence of HPWMoV in Eurasia, and EFSA considers it a potential quarantine pest for the EU [80]. Notably, both research groups used the same set of primers targeting the HPWMoV nucleocapsid region developed by Lebas et al. [81], which is located within genomic RNA 3 characterized by two heterogenous variants [45]. The higher specificity afforded by the RT-PCR assay may hinder the detection of divergent strains, and the lower sensitivity of ELISA could tolerate substitutions at key positions, making it a useful tool for the first detection of viruses in a new location. While the recent introductions in Iran and Iraq are troubling, the spread of the virus has not reached the levels observed in North America and could be avoided through appropriate quarantine measures.

WSMV is one of the most common viruses in wheat-growing regions on six continents. Its topology consists of four main clades: A (Mexico), B (Europe, Russia, and East Asia), C (Iran and West Asia), and D (the USA, Argentina, Brazil, Australia, Turkey, and Canada) [28,49]. It has been suggested that the American isolates of WSMV form a population separate from Eurasian isolates, and Turkish isolates are known to be closer to American than Eurasian ones [22,33]. These results indicate that there has been a separation of American and Turkish isolates from isolates from Mexico, Central Europe, Russia, and Iran. The 23 isolates found in Kazakhstan were grouped into one cluster with European isolates, placing them inside Clade B. However, isolate KZ57 (Kostanay) clustered away from Clade B and shared a high-support (bootstrap value 100) common ancestor with Australian and American sequences of Clade D as well as several Iranian isolates. It also showed evidence of recombination according to RDP5, supporting the previous findings that the 3′ end of the CP coding region is a common recombination spot [82]. Yet, only limited conclusions can be drawn from the sequence analysis of one of the most variable fragments of a viral genome, and whole genome sequencing can shed more light on potential sources of infection.

Another important factor is that seed transmission, while possible, is infrequent: both 0.5–1.5% [54] and up to 13% in susceptible cultivars have been reported [55]. Thus, it is unlikely that the incidence of 21 viral infections (as in location 7) is the product of infected international germplasm. WSMV is primarily transmitted via wheat curl mites (WCM), which can acquire the virus after a 15 min feeding session [83] and spread it around the field. Currently, no reports or comprehensive studies exist on WCM of Kazakhstan, so further research of the topic of the wheat streak mosaic should be based upon a deeper understanding of the WCM distribution within the grain growing locations.

According to the data from the State Revenue Committee of the Ministry of Finance of the Republic of Kazakhstan, the predominant direction of foreign wheat trade is Europe [84]. Thus, the genetic relationship among isolates could be explained by the flow of the viral genetic pool via wheat seeds accompanied by insects. However, the available data do not define the exact proportion of flow or the time. The primary testing of imported wheat seeds and the correlation between the source of infection and cultivars could be more useful in the prediction of the infection’s origin.

Throughout the years, BSMV has been detected in North America, Europe, Asia, and North Africa [21,85]. BSMV sequences from Kazakhstan showed a high nucleotide identity with those from Europe; however, the low diversity of available isolates makes it difficult to draw conclusions about the global spread of the virus. While the main source of barley seed imports to Kazakhstan is Iran [84], the detected BSMV isolates were found in proximity to European isolates. Despite the lack of sequence data from other regions, particularly West Asia and Iran, it could be assumed that the virus was introduced alongside the wheat seed material imported from Europe. However, the well-supported cluster composed of isolates from China and Egypt could represent the spot where Iranian isolates could be located within this phylogenetic tree.

Thus, it is important to establish strict control over grain imports to avoid new infection transfers and to take appropriate epidemiological measures to prevent the further spread of isolates already present in the country. Factors such as the possible lack of spatial isolation of winter and spring crops and grasses, the late sowing of spring and early sowing of winter crops, and the presence of weeds and vector pests could lead to the preservation and spread of these viruses. While BSMV can be managed by seed testing and the timely isolation of the infected plants, WSMV is more difficult to control because of the mobility of vector mites. The principal method is to control the vector population by disrupting their life cycle [86]. Therefore, it is necessary to destroy the residual harvest in a timely manner after planting to effectively combat wheat mites. In addition, it is necessary to till the soil and destroy the surrounding weeds with herbicides, which will help reduce the sources of viral inoculum and the mite population. Crop rotation and methods of sowing crops at the beginning/end of the season can also reduce the vector populations and prevent the spread of viral infections.

## 5. Conclusions

Despite the crucial role of wheat in the economy and food safety of Kazakhstan, the lack of epidemiological studies on plant viruses leaves wheat production in the country vulnerable to viral infections. Molecular genetic techniques are among the key modern methods in the epidemiology, ecology, and biology of viral infections. Methods including LAMP, RT-PCR, RT-qPCR, and sequencing allow prompt virus diagnosis, thereby facilitating the management of plant diseases. The present study will contribute to the development and implementation of new effective and long-term strategies for the control of cereal viruses in the country, as well as providing a better understanding of the genetic variability in local populations of WSMV and BSMV. Further work will involve improving sequencing quality and targeting whole genome sequences as well as deepening the understanding of local wheat curl mite dispersal.

## Figures and Tables

**Figure 2 viruses-16-00096-f002:**
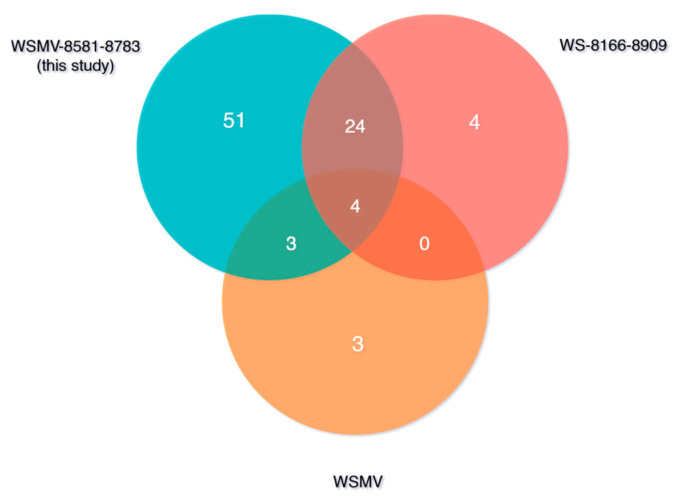
Overlap in amplicons produced by the three sets of WSMV primers. WSMV-8156-8783 (this study) was the only set to amplify 51 of the isolates analyzed; another 24 were also detected by WS-8166-8909 [62] and 3 by WSMV [43]. Only four isolates amplified all three sets of primers.

**Figure 3 viruses-16-00096-f003:**
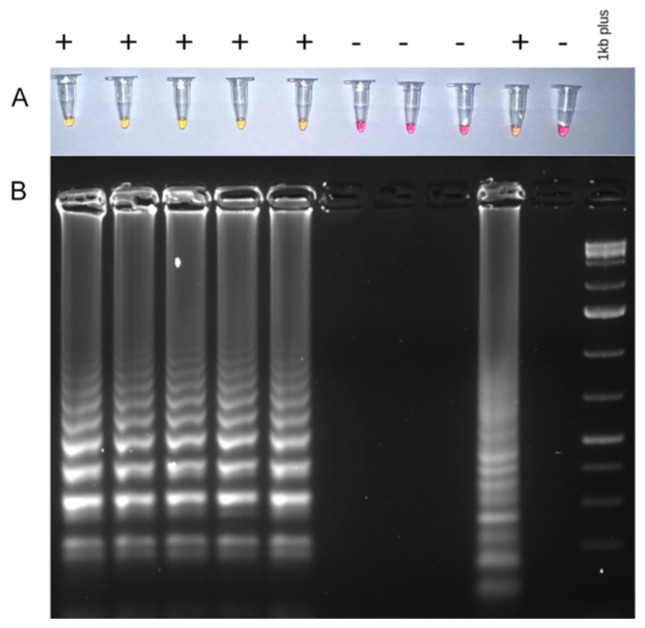
Wheat streak mosaic virus (WSMV) detection using LAMP assay. (**A**) Amplification analysis using the colorimetric method; pH-sensitive indicator phenol red changes the color from red to yellow in positive samples. (**B**) LAMP products separated on 2% agarose gel.

**Figure 4 viruses-16-00096-f004:**
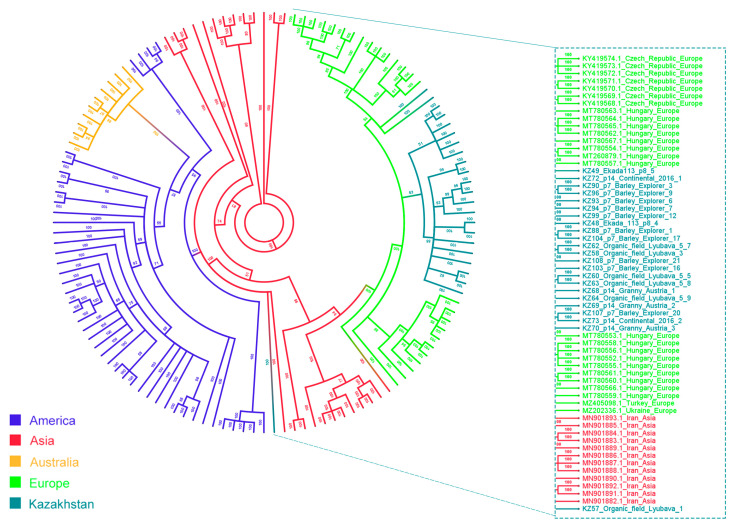
Cladogram constructed from all publicly available coat protein sequences of WSMV using the Bayesian algorithm, based on 950 bp sequences of the coat protein coding region. Branches and taxa are according to their continent of origin.

**Figure 5 viruses-16-00096-f005:**
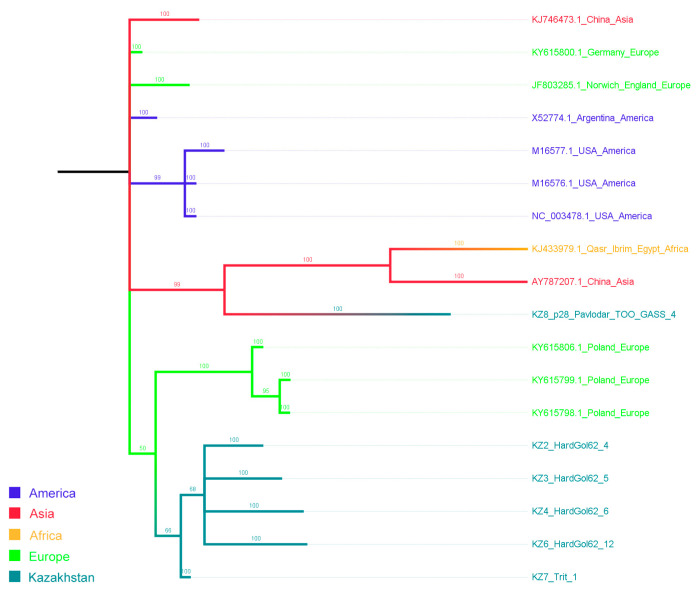
Phylogenetic tree constructed from all publicly available coat protein sequences of BSMV using the ML algorithm, based on 460 bp sequences of the γB protein coding region. Branches and taxa are according to their continent of origin.

**Table 1 viruses-16-00096-t001:** Primers for the detection of wheat streak mosaic virus (WSMV), barley stripe mosaic virus (BSMV), triticum mosaic virus (TriMV), and *Emaravirus tritici* (previously HPWMoV).

Primer Name	Target Virus	Target Region	Primer Sequence	T_m_ (°C)	Product Size (bp)	Source
WSMV-8581F	WSMV	coat protein	CGACAAGATAAAGCCTGAAT	54	274	This study
WSMV-8783R			AGTGCTCTGTTCTCTGG			
BSMV-2763F	BSMV	γB protein	AGAAGAAGATGCAGGAGCTGA	58	134	This study
BSMV-2868R			AAGAATCATCACATCCAACAG			
WS-8166F	WSMV	coat protein	GAGAGCAATACTGCGTGTACG	45	750	[62]
WS-8909R			GCATAATGGCTCGAAGTGATG			
WSMV	WSMV	coat protein	GTTGGGAGGCTTAATTGAAGTG	55	720	[43]
			CAGCCATTACTCGTGTTATCCA			
TGB2F	BSMV	γB protein	GGATGAAGACCACAGTTGGTTC	55	397	[59]
TGB2R			CTAGCCAATATCGCATAGTAATG			
TriMV	TriMV	PIPO region	CTTAAGCACATGTTACAATC	45	1200	[43]
			GTCCCTGATAACTAATTCTA			
WMoV	HPWMoV	RNA3	GTTCCAATTCCTGTGCTTGATCTGTC	45	490	[43]
			AACAATGACATAGCAATTACCTCAGCA			

**Table 2 viruses-16-00096-t002:** Primer sets for the detection of WSMV using LAMP assay.

Set	Protein	Primer	Sequence (5′-3′)
1	CP	F3	GCAGCAACTGATGCTGTCT
		B3	CTTGACCAGTCTTGGCTCC
		FIP	GCCGTGCTTGCACTCAGACTCGCAAATGCGGAAGTGGCA
		BIP	ACCATCAGGATCAGGTTCCGGATGACCGACACGTTGCTAGA
		LB	GCGGGTGGATCAGGTTCAG
		LF	GCGTGCTTCCACTACTCG
2	CP	F3	CGAGTGAGCAGCAACTGA
		B3	GACACGTTGCTAGACTGTGT
		FIP	GACTCGACTGCGTGCTTCCACTCTGGCAGCAGCAAATGC
		BIP	CAAGCACGGCAAGCGGATCAGATCCTGAACCTGATCCACC
		LB	TCACCATCAGGATCAGGTTCCG
3	NIb	F3	GCATACTACAAGGACGTGTT
		B3	TTTCTTCCCTGAATATAACGC
		FIP	TAGTTGCAACCACTGCCTTCGGTACAACAAAGATATAATTGTCGGA
		BIP	GAGATAGCCGGGTTCTCAAAGGGCATCCAGATTCAAGTCGTT
		LB	AAAGCACACTTCATCAGTTCC
		LF	GACAATTTCAGCCCATCTTTGA

**Table 3 viruses-16-00096-t003:** Primers for the sequencing of long reads of the WSMV CP and BSMV γb protein.

Target Gene	Forward Primer (5′→3′)	Reverse Primer (5′→3′)
WSMV coat protein	TCGAGTAGTGGAAGCACTCAGTC	CATGACAGATACGTTATTAGATTG
	TCATGCGCGGTGCAGATGACACA	ATGACGTGAGTTGTCCTCATTAG
BSMV γB	ATGATGGCTACTTTCTCTTGTGT	TTACAACTTAGAAACGGAAGAAG
	TTCTTGTTTCAGAATATACATGC	GCTAAGCTTGAAAGTGAGGTTAA

## Data Availability

Raw data are available in the Appendix A.

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
