# Peer review of "Distribution of Wheat-Infecting Viruses and Genetic Variability of Wheat Streak Mosaic Virus and Barley Stripe Mosaic Virus in Kazakhstan"

_viruses, 2024, doi:10.3390/v16010096_

Round 1
Reviewer 1 Report
Comments and Suggestions for Authors
The authors surveyed wheat fields for WSMV, BSMV, TriMV, and HPWMoV in Kazakhstan, and found only WSMV and BSMV. The authors used RT-PCR and LAMP-mediated detection methods for wheat viruses, and found 82 and 19 samples for WSMV and BSMV, respectively, in 256 field-collected samples. The quality and quantity of work presented in this manuscript are not sufficient for a publication in Viruses. The authors performed superficial studies on field surveys, detection, sequencing, and phylogenetic analyses. They should have performed in-depth experiments on the development of LAMP assay for field-based detection of wheat viruses, and sequence of complete genomes of WSMV and BSMV for a few isolates. A manuscript with a first-time report on virus distribution and diversity without an in-depth study is not enough for publication. This manuscript is suitable for a regional journal. This manuscript needs moderate edits for English.
Some select specific comments:
L27-29: Needs a reference. Maybe Phytopathology 113: 117-141.
L55: Change High Plain mosaic virus to High Plains virus.
L56: Reference 24 is not appropriate. Cite Tatineni et al. 2014. J Virology 88: 11834-11845.
L166-168: No in-depth study on recombination.
Comments on the Quality of English Language
Needs moderate edits for English for grammar and sentence structure.
Reviewer 2 Report
Comments and Suggestions for Authors
In this manuscript Kapytina and colleagues investigated the presence of WSMV, BSMV and two other viruses (TriMV and HPWMoV) in wheat and barley samples In Kazahstan. They optimised primers for RT-PCR and LAMP tests. Amplified partial CP of WSMV and BSMV and phylogenetically analysed the sequenced products.
Until now there is no sequence data about cereal viruses from Kazahstan why I think this report is important, but I would suggest to publish it as a short report.
I found several shortcomings of the manuscript and detailed here my suggestions:
from line 25: The introduction does not contain information about how important cereals are, how large is the area of its cultivation in Kazahstan. If you have any data about the known yield loss because of viruses it would be nice to add.
from line 32: The most important wheat and barley infecting viruses are introduced briefly. In the GenBank there are 157 and 63 viruses that infects wheat and barley, respectively. I think it should be at least state why the chosen 4 are discussed in this research. Wheat dwarf virus is an important virus, present in Europe causing problems. Isn’t it present in Kazahstan? If yes, it should be introduced, if not this should be mentioned.
from line 32: The introduction of viruses is not uniform; genome organization is only described for BSMV and WSMV. For the virus description the same order (alphabetical, or based on importance) would be nice to keep, from the listing till the more detailed introduction.
As important part of the manuscript is about the improvement of BSMV and WSMV diagnostics, a paragraph about their possible available diagnostics would be nice to add.
line 90: Table A1 contains information about the samples, but only in case of when virus infection was detected. This is why I would suggest to rename its title. My suggestion is: Sample collection places where WSMV or BSMV infection was detected.
Fig1: I would suggest not to include Fig1C left panel. and would include Fig 1C right two panel (results of the RT-PCR) on a different Figure.
line 96-97: Was the RT done by a mixture of oligodT and Random primer, as the text suggest?
Table 1 contains information about the used primers, but in a very mixed-up style. I would suggest to insert a new column as first (name of the virus), listing all of the used primer pairs for all of the viruses and include information of the amplified part also as an extra column. Only 1 name is listed, while there are two primers. Primers should have their own name as they can be used independently, why in the Table 1 two names should be referred, this should be also clarified.
line110: It is written that if the samples were negative for all viruses. Names of all of the tested viruses should be listed to be clear.
line 114: New primers for WSMV and BYSM detection were designed using Primer3, I missed the information what sequences were used as a base of the design. Please describe this part in more detail.
Line180-182. I don’t think that the error of the RT enzyme can cause problem in the RT-PCR detection. These errors are happening randomly, so as a sum, if the primers can anneal perfectly this would be masked.
Figure 2. I don’t understand how this figure shows the performance of the primers? WSMV-CP-274 amplified 92% of the isolates analysed, but it is about the half of the circle. I think for the performance data a table is enough, or if you want a diagram a column diagram would be better. How do you know that these primers amplified 92% of all isolates If only RT-PCR was used there would be a chance to lose a variant because of primer problem. To refer % is always misleading, if you are not sure about the 100%. Referring numbers instead of % would be more convincing.
line193. I guess you ordered the LAMP primers, so please delete synthetised – it should suggest that you synthetised them.
line194. With LAMP, you can see only the increasing turbidity, color change and lots of bands on the gel, why you think that nonspecific amplification happened in case of two sets? I cannot find the numbers what referred false negative or positive on the RT-PCR result. In theory as the LAMP primers were designed to amplify another region of the virus than the RT-PCR primers, it is possible that they can amplify a strain which has a mutation in the PCR primer annealing part of the viral genome. I think a table about the numbers how many samples were detected positive using RT-PCR with the optimised primers and with LAMP would be very nice to see.
line200: as there are contradictions between the detection techniques, I don’t think that this sentence is true, or if yes only with some conditions.
from line 205 Chapter 3.2.: I think it is a good idea to carry out phylogenetic analysis, but sequencing of the amplified part of the viruses won’t tell anything about the specificity of the designed primers. As it cannot tell result about strains which were failed to be amplified by the used primers.
Fig4.: I think for the phylogenetic analysis the 274nt long part of a Potyvirus is very misleading, so I suggest not to include Fig4/B. and similarly leave out Fig5/B where a very short 134bp long sequence is used for phylogenetic comparison, what is absolutely not conclusive.
On Fig4/A panel, it is clearly visible that the Kazah variants positioned on a branch, where all of the European variants are, however from the detailed part information about half of these closest European variants are missing. (I suggest the include here all green isolates).
line243-249. I don’t think that the recombination analysis is a good idea for these short sequences either. Indeed, in any experiments where a sequence is not cloned as a single molecule there is a possibility that the sample contains slightly different strains, which were mixed up during the the novo assembly, which could be misdiagnosed as a recombination. I would leave out this paragraph.
Based on the above critical points: I found some problematic points in the discussion which I suggest to rewrite when the main result section was revised.
line 264-275: TriMV and HPWMoV were detected in the cited papers using DAS-ELISA, what is not as sensitive as RT-PCR, but has less problem with the SNPs. To reveal the presence of these viruses in Kazahstan az ELISA based survey would be a good choice.
line: 288-290: I would leave out the recombination analysis.
line 283-286: Discussing the phylogenetic analysis, I would not cite the number of nucleotide differences, because the are correlated with the length of the investigated viral sequence (for a short fragment 12 SNP can be a lot, while on the whole virus level is very low. It depends also which part of the viral genome we compare).
line 291-298: To discuss the possible source of the infection I don’t think that investigation of more isolates would be the best way the found out the origin. Moreover, to check the original import source, and correlate the bred with the source of the infection would be better.
As a summary I found this short report interesting, but I think it needs a careful revision and can be published only when the cited issues are solved and corrected.
Reviewer 3 Report
Comments and Suggestions for Authors
In every aspects this is a correct work and manuscript as well.
A few things that should be added to the manuscript: In the introduction part I would add the transmission of wheat streak mosaic virus similarly than barley stripe mosaic virus. WSMV is mite transmitted virus the seed transmission has a very low frequency as 0.5 to 1.5%.
How to imagine the European origin of Kazakh WSMV isolates, or it can be explained by seed transmission, which is low but exists?
I suggest to remove sentence connected tu MMLV RT in line 180-182. In my opinion, the fidelity of RT does not cause amplification problems.
For Figure 3, parts A and B need to be aligned better.
It is not clear what percentage of symptomatic and asymptomatic samples gave a positive result, in how many samples were they able to detect one of the tested viruses.
How do you explain false positive and false negative results using LAMP detection at FigS1?
Round 2
Reviewer 1 Report
Comments and Suggestions for Authors
The manuscript improved, though the quality of research may not fit for Viruses. I recommend for its acceptance.
Comments on the Quality of English LanguageNeeds minor edits to English.
Reviewer 2 Report
Comments and Suggestions for Authors
Dear Authors,
Many thanks for answering my questions and comments.
The revision was done based on my comments and my questions were answered, I appreciate that.
I think that the manuscript improved and I believe that in this revised form it can be accepted for publication in the MDPI Viruses.